# High-Throughput Identification of MiR-145 Targets in Human Articular Chondrocytes

**DOI:** 10.3390/life10050058

**Published:** 2020-05-11

**Authors:** Aida Martinez-Sanchez, Stefano Lazzarano, Eshita Sharma, Helen Lockstone, Christopher L. Murphy

**Affiliations:** 1Kennedy Institute of Rheumatology, University of Oxford, Oxford OX3 7FY, UK; stefano.lazzarano@gmail.com; 2Wellcome Trust Centre for Human Genetics, University of Oxford, Oxford OX3 7BN, UK; esh@well.ox.ac.uk (E.S.); hel23@well.ox.ac.uk (H.L.)

**Keywords:** miRNAs, chondrocyte, cartilage, RNA-immunoprecipitation

## Abstract

MicroRNAs (miRNAs) play key roles in cartilage development and homeostasis and are dysregulated in osteoarthritis. MiR-145 modulation induces profound changes in the human articular chondrocyte (HAC) phenotype, partially through direct repression of *SOX9*. Since miRNAs can simultaneously silence multiple targets, we aimed to identify the whole targetome of miR-145 in HACs, critical if miR-145 is to be considered a target for cartilage repair. We performed RIP-seq (RNA-immunoprecipitation and high-throughput sequencing) of miRISC (miRNA-induced silencing complex) in HACs overexpressing miR-145 to identify miR-145 direct targets and used cWords to assess enrichment of miR-145 seed matches in the identified targets. Further validations were performed by RT-qPCR, Western immunoblot, and luciferase assays. MiR-145 affects the expression of over 350 genes and directly targets more than 50 mRNAs through the 3′UTR or, more commonly, the coding region. MiR-145 targets DUSP6, involved in cartilage organization and development, at the translational level. DUSP6 depletion leads to MMP13 upregulation, suggesting a contribution towards the effect of miR-145 on MMP13 expression. In conclusion, miR-145 directly targets several genes involved in the expression of the extracellular matrix and inflammation in primary chondrocytes. Thus, we propose miR-145 as an important regulator of chondrocyte function and a new target for cartilage repair.

## 1. Introduction

The articular chondrocyte is the only cell type present in the articular cartilage, solely responsible for synthetizing the components of the extracellular matrix (ECM) that allows an almost friction-free, painless movement of the joint. When the articular chondrocyte function is altered, the structure of the ECM can be compromised leading to the development of osteoarthritis (OA) [1]. Local inflammation within the joint may have a detrimental effect in chondrocyte function and further contribute to the development of OA [2]. Currently, there is no cure for OA, which affects over 250 million people worldwide, (WHO http://bjdonline.org/), and most common treatments can only aim to pain reduction.

A normal healthy articular chondrocyte maintains the equilibrium between anabolic molecules, such as collagen type II (COL2A1)—a main component of the ECM—and catabolic enzymes such as the collagen-degrading metalloproteinase MMP13 [1,3]. Understanding the mechanisms underlying the human articular chondrocyte (HAC) function is indispensable to develop therapies aiming to restore the capacity of the HAC to produce a balanced ECM. Failure to develop drugs to stop the development of the disease is partially due to the lack of a comprehensive knowledge of the factors leading to cartilage degeneration and to the poor intrinsic repair capacity of articular chondrocytes [4]. Moreover, joint damage often acts to alter chondrocyte homeostasis of the remaining viable chondrocytes causing further degeneration and/or hypertrophy and matrix calcification over time [5]. Recent research has identified mesenchymal stem cells with chondrogenic potential as a promising source for tissue engineering and replacement, which may suffer less hypertrophy/dedifferentiation following transplantation [6].

MicroRNAs (miRNAs) are small RNAs that silence gene expression post-transcriptionally [7]. Over two thousand miRNAs are expressed in humans [8] that play regulatory roles in most biological processes, development, and diseases [9,10,11]. Animal miRNAs recognize the target mRNAs via partial sequence complementarity and recruit the miRNA-induced silencing complex (miRISC). The MiRISC action results in decreased protein production as a consequence of reduced mRNA translation and/or increased mRNA degradation [7].

MiRNAs have been widely shown to not only regulate cartilage development but also to modulate the normal function of the adult articular chondrocyte [12,13,14]. We have previously identified miR-145 to profoundly impact the expression of essential components of the ECM, such as ACAN, COL2A1, or MMP13, at least partially by directly targeting the master regulator, transcription factor, SOX9 [15]. Interestingly, miR-145 levels are increased in OA chondrocytes [16] raising the possibility of inhibition of miR-145 as a promising approach for cartilage repair. Nevertheless, a single miRNA typically targets dozens to hundreds of mRNAs simultaneously [7], which could result in unpredicted phenotypes upon manipulation. Therefore, if miR-145 is going to be considered as a target for cartilage repair, it is fundamental to determine its whole targetome in HACs. Current available computational methods for miRNA target determination rely in the extent of the complementarity between the seed region of the miRNA (six nucleotides positions 2 ± 7) and the 3′UTR of the mRNAs, which results in hundreds of predictions, most of which do not represent bona-fide interactions [17]. Additionally, the efficiency of these tools tends to be poor for the detection of nonconserved binding sites, binding sites with poor pairing or for those located outside the 3′UTR of the mRNAs [17]. Moreover, computational approaches do not take into account the possibility of tissue-specific interactions [17,18].

Therefore, we aimed to determine miR-145 targets in HACs using an unbiased, high-throughput experimental approach capable of detecting actual miRNA–mRNA interactions [18]. Accordingly, in the present study we performed RNA/miRISC-immunoprecipitation experiments [19,20,21] followed by high-throughput sequencing (RIP-seq) in primary HACs isolated from healthy human cartilage and previously transfected with a miR-145 mimic. Using this method, we detected 59 mRNAs enriched in miRISC-immunoprecipitation upon miR-145 overexpression, indicating direct miRNA–mRNA interactions. Reporter assays confirmed that miR-145 silences the expression of several of these genes through sequences present in their 3′ untranslated region (3′UTR) or coding sequence (CDS). One such identified mRNA, DUSP6, had previously been reported to regulate cartilage function and inflammation of different tissues. The predicted miR-145 binding site in DUSP6 was validated by reporter assays and, as expected, DUSP6 protein levels were modulated in primary HACs upon miR-145 overexpression/inhibition. Both miR-145- and siRNA-mediated downregulation of DUSP6 resulted in increased expression of the catabolic gene MMP13, indicating that DUSP6 is at least partially responsible for the effect that miR-145 exerts in the MMP13 gene expression.

## 2. Materials and Methods

### 2.1. Cell Culture and Transfection

Following local ethics committee guidelines and after informed consent, healthy human articular cartilage was extracted and HACs were isolated from the femoral condyle and tibia plateau following amputation due to sarcomas not involving the join space. Cartilage samples were collected on the day of surgery, diced into 1–2 mm pieces and digested with 1.5 mg/mL of collagenase type II (Worthington) in DMEM—10% FCS during 18 h at 37 °C. Isolated human articular chondrocytes (HACs) were then strained, pelleted, washed, and seeded at a density of 1.2 × 10^4^ cells/cm^2^ as previously described [15]. Primary cells (passage 0—P0) were harvested after 5–7 days of culture or subcultured for a further 5–7 days and passaged one, two, or three times (P1–P3). HACs, as well as HeLa cells, were kept in DMEM supplemented with 10% FCS in a humidified 5% CO_2_ atmosphere at 37 °C under normal (20%) or hypoxic (1%) oxygen tension, when indicated.

### 2.2. Plasmids

The sequence containing three consecutive perfect matches for hsa-miR-145-5p (AGGGATTCCTGGGAAAACTGGACAGCGTAGGGATTCCTGGGAAAACTGGACAGCGTAGGGATTCCTGGGAAAACTGGAC) was excised from the previously produced pSG5-Luc-3xmiR-145 ^15^ and subcloned between XhoI–XbaI restriction sites in a pMirGlo dual-luciferase miRNA target expression vector (Promega) to generate pMiRGlo 3xmiR-145.

MOK 3′UTR, MOK CDS, CYR61 3′UTR, HSBP1 3′UTR, RTKN 3′UTR, CTBP1 3′UTR, CTBP1 CDS, FKBP2 CDS, HIST1H4E full-length, GLT8D1 3′UTR, RAB7A 3′UTR, RAB7A CDS, PLOD1 3′UTR, SEC11A 3′UTR, FN3K CDS, UXS1 3′UTR, and DUSP6 3′UTR were amplified by PCR using specific oligonucleotides (sequences available upon request) from HACS cDNA, inserted into the pGEM-T vector (Promega) and subcloned into pMiRGlo.

Three-point mutations in the predicted miR-145 seed binding site in DUSP6 were introduced in pGEM-T-DUSP6 3′UTR using a Phusion^®^ site-directed mutagenesis kit (NEB) and the mutagenic primers: Forward: 5’-TGTGAGCATGGGTACCCATT-3’ and reverse 5’-CACACACACTTCGTCTTTTATACAAA-3’. Mutated DUSP6 3’UTR was subcloned into pMiRGlo. All the constructs were verified by sequencing.

### 2.3. Transfection of HACs

HACs were seeded at 1 × 10^4^ cells/cm^2^ and, 12–24 h later, transfected using Lipofectamine 2000 (Thermo Fisher Scientific) and 5 nM control or miR-145 mimics (AM17110, AM17100-PM11480, respectively, Thermo Fisher Scientific), 10–50 nM control or hsa-miR-145 miRCURY LNA Power inhibitors (Cat No. 339136, YI00199006, YI04102423, respectively Qiagen, Netherlands) or 5 nM control or DUSP6 siRNA (On-Target Plus, J-003964-06 (DUSP6 siRNA #1), J-003964-07 (DUSP6 siRNA #2), D-001810-01 (Control), Horizon Discovery) following the manufacturer’s instructions and as previously described [22]. After transfection, the medium was replaced by a pre-equilibrated DMEM (in 20% or 1% oxygen tension) containing 10% FCS and antibiotics. Cells were kept at the appropriate oxygen tension for a further 24–48 h.

HeLa cells were seeded at 60% confluence and, 12–24 h later, transfected with 100 ng of plasmidic DNA together with 5 nM control or miR-145 mimics using Lipofectamine 2000 (Thermo Fisher Scientific). After 24 h, cells were lysed and luciferase activity determined with the Dual-Glo luciferase assay system (Promega, UK), following the manufacturer’s instructions.

### 2.4. Ribonucleoprotein Immunoprecipitation, mRNA–miRNA Isolation and miRNA Reverse Tanscription (RT), and Real-Time PCR (qPCR)

Ribonucleoprotein immunoprecipitation was performed as described previously [21]. Briefly, 30 × 10^6^ P3 chondrocytes from three different patients (19 yo. female; 8 yo. male; 28 yo. male) were transfected with miR-145 or control mimics and, 24 h later, lysed in an equal volume (300–400 μL) of an ice cold polysome lysis buffer (5 mM MgCl2, 100 mM KCl, 10 mM Hepes, pH 7.0, and 0.5% Nonidet P-40) with freshly added 1 mM DTT, 100 units/mL RNase OUT (Invitrogen), and complete mini EDTA-free protease inhibitors mixture (Sigma-Aldrich) during 5 min, snap frozen, and kept at −80 °C for at least 18 h. Samples were then centrifuged at 14,000 g at 4 °C for 10 min and protein concentration was determined by the Bradford assay 2.

The supernatant containing 6 mg of total protein per condition was mixed with 2.7 mL of an ice cold NT2 buffer (50 mM Tris, pH 7.4, 150 mM NaCl, 1 mM MgCl2, 0.05% Nonidet P-40) containing freshly added 200 units/mL RNase OUT (Thermo Fisher Scientific), 0.5% vanadyl ribonucleoside (Thermo Fisher Scientific), 1 mM DTT, 15 mM EDTA, and 90 µl Sepharose G beads (Abcam, Cambridge, UK) precoated for at least 4 h with 6 μg of mouse antihuman Ago2 (clone2E12-1C9, Abnova). Incubation was carried out overnight at 4 °C on a rocking platform. On the following day, beads were washed six times with a 3 mL ice cold NT2 buffer and 5% of the beads were separated for Western blotting. The rest of the beads were incubated in a 1% SDS-NT2 buffer during 30 min at 50 °C in order to elute the bound RNA that was isolated using TRIzol and glycogen (Invitrogen, UK) as a carrier in the precipitation step. The mouse IgG1 isotype control (Abcam) was used as a negative control for the immunoprecipitation procedure.

In parallel, total RNA was extracted using TRIzol from a 10% volume of the samples submitted to immunoprecipitation, prior to incubation with the Ago2 antibody.

For miRNA detection, 5% of the recovered RNA was reverse transcribed using the TaqMan miRNA RT kit (Thermo Fisher Scientific) and miR-145, miR-140, and RNU24-specific primers (Thermo Fisher Scientific). QPCR was performed with a TaqMan PCR master mix (Thermo Fisher Scientific) and the appropriate TaqMan probes (Thermo Fisher Scientific). The remaining RNA was treated with DNAse I (Thermo Fisher Scientific) and sent for high-throughput sequencing.

### 2.5. mRNA-Seq Library Construction, Sequencing, and Analysis

RNA quantity and integrity were assessed using the Quant-IT RiboGreen Kit (Invitrogen) and Agilent Tapestation 2200 R6K. mRNA enrichment was achieved from 400–900 ng of total RNA using a Magnetic mRNA Isolation Kit (NEB, UK). Generation of double stranded cDNA and library construction were performed using the NEBNext^®^ mRNA Sample Prep Reagent Set 1 (NEB). The ligation of adapters was performed using adapters prepared at the WTCHG according to the Illumina design (Multiplexing Sample Preparation Oligonucleotide Kit, NEB, UK). Each library was subsequently size selected with two AMpure Bead bindings. The following custom primers were used for PCR enrichment: Multiplex PCR primer 1.0: 5’-AATGATACGGCGACCACCGAGATCTACACTCTTTCCCTACACGACGCTCTTCCGATCT-3’. Index primer: 5’-CAAGCAGAAGACGGCATACGAGAT[INDEX]CAGTGACTGGAGTTCAGACGTGTGCTCTTCCGATCT-3’. Indices were according to the eight bases tags developed by WTCHG [23]. The following modifications to the described workflow were applied to RNA recovered after ribonucleoprotein immunoprecipitation. Firstly, no poly(A) selection or ribodepletion was applied. Secondly, the number of PCR cycles was increased from 12 to 15. Amplified libraries were analyzed for size distribution using the Agilent Tapestation 2200 D1K. Libraries were quantified by quantitative RT-PCR using the Agilent qPCR Library Quantification Kit and a MX3005P instrument (Agilent, Germany) and relative volumes were pooled accordingly. Finally, a second RT-qPCR was performed to measure the relative concentration of the pool compared to a previously sequenced mRNA library in order to determine the volume to use for sequencing. Sequencing was performed as a 50 bp paired end read on a HiSeq2000 according to Illumina specifications.

FASTQ files were generated for each sample and initial data quality checks of the raw sequence data were performed. Reads were then mapped to the human genome (GRCh37) using TopHat [24]. HT-seq [25] was used to summarize the mapped reads into a gene count table, the final version of which contained one column per sample. Ensembl (v65) annotations were used to define gene models. All subsequent data analysis was performed in R using relevant Bioconductor packages [26].

Total RNA profiles were consistent and generated ~20 million reads mapping to Ensembl genes per sample. IP samples generated 2–3 million reads per sample and one potential outlier sample was observed on clustering and PCA plots (Appendix A). The R package ‘DESeq’ [27] was used to normalize the samples for library depth and perform a paired analysis to find differentially expressed genes between the samples overexpressing miR-145 and controls. Total RNA and IP samples were treated as separate datasets. The two resulting gene lists were compared to identify genes that were significantly upregulated in the IP samples while being significantly downregulated or unchanged (not significant, FDR > 0.05) in the total RNA samples as potential targets of miR-145. For the analysis performed with IP samples from only two patients, we investigated the effect of the outlier and the final IP list was generated excluding both samples from the patient (donor 3) with the outlier sample (miR-145 donor 3). Genes were filtered by expression levels and those with log2CPM > 2 from total RNA and IP samples were independently ranked by fold change of expression and subjected to the motif discovery method cWords in two separate analysis per list, one for enrichment in the 3′UTRs and the other in the CDS [28].

### 2.6. mRNA Reverse Transcription (RT) and Real-Time PCR (qPCR)

Total RNA was extracted from P0–P3 HACs using TRIzol according to the manufacturer’s instructions. An amount of 200–500 ng of RNA was reverse transcribed using the High Capacity cDNA RT kit (Thermo Fisher Scientific). QPCR followed, using a SYBR Green PCR master mix (Thermo Fisher Scientific) and specific primers (sequences available upon request). The age and gender of the HACs donors used for each experiment are indicated in the figure legends.

### 2.7. Western Blotting

P0–P3 HACs were cultured as monolayers and transfected 36–48 h before lysis. When appropriate, HACs were stimulated with a 25 nM human FGF-2 (Prepotech, UK). The age and gender of the HACs donors used for each experiment are indicated in the figure legends. HACs were lysed in a radio-immunoprecipitation assay buffer (RIPA—150 mM NaCl, 10 mM Tris, pH 7.2, 5 mM EDTA, 1% Triton X-100, 0.1% SDS, 1% deoxycholic acid) and 10–30 μg total protein extract subjected to SDS-polyacrilamide gel electrophoresis and transferred to PVDF membranes (Merck-Millipore, Germany). Mouse monoclonal anti-Ago2 (clone2E12-1C9, Abnova, 1:1000 dilution), rabbit monoclonal anti-DUSP6 (EPR129Y, Abcam, 1:1000 dilution), rabbit monoclonal anti-phospho-ERK (Cat No. 05-797R, Sigma-Aldrich, 1:1000 dilution), rabbit polyclonal anti-ERK (Cat No. sc-292838, SantaCruz Biotechnologies, 1:500 dilution), mouse monoclonal anti-Tubulin (clone B-5-1-2, Sigma-Aldrich, 1:5000 dilution), HRP-conjugated anti-mouse IgG secondary (Cat No. NA931, GE Healthcare, 1:5000 dilution), and HRP-conjugated anti-rabbit IgG secondary (Cat No. NA934, GE Healthcare, 1:5000 dilution), antibodies were used. Proteins were visualized by ECL fluorography (GE Healthcare, Germany).

### 2.8. Statistical Analysis

GraphPad Prism 6.0 was used for statistical analysis. Statistical significance was evaluated by the two tailed paired student’s *t*-test or two-way ANOVA and Fisher’s LSD test for multiple comparisons. All data are shown as mean ± standard error of the mean (SEM). *p*-values < 0.05 were considered statistically significant.

## 3. Results

As previously mentioned, miRNAs direct miRISC to the target mRNAs which, as a consequence, are less efficiently translated and/or degraded. Thus, regardless of the subsequent mechanism of repression, an mRNA targeted by miRNAs will be bound to the miRISC. Consequently, overexpression of a specific miRNA is expected to increase the binding of its mRNA targets to miRISC. MiRISC can be immunoprecipitated using antibodies against miRISC proteins such as Argonaute 2 (AGO2), a key component of the complex.

In order to determine miR-145 direct targets in human chondrocytes we performed mRNA/miRISC-immunoprecipitation followed by high-throughput sequencing of the RNA (RIP-seq) with an anti-Ago2 antibody and a lysate from P3 HACs previously transfected with a miR-145 mimic or a nontargeting control. In parallel, to assess the overall impact of miR-145 overexpression in gene expression in HACs, we sequenced total mRNA (T-RNA) of HACs transfected with miR-145 or control mimics. Upon miR-145 overexpression, miR-145 target mRNAs are expected to be enriched in Ago2 precipitates while their levels in the total RNA samples should remain unchanged or be reduced (depending on whether miR-145 suppresses target mRNA translation or mRNA stability, respectively).

The specificity of Ago2-immunoprecipitation was verified by Western blotting (Figure 1A) and hundreds of miRNAs were detected in Ago2 immunoprecipitates in comparison with an IgG control (data not shown). The amount of miR-145 associated with Ago2 was 40-fold higher in cells overexpressing miR-145 (Figure 1B). Immunoprecipitated RNA (IP-RNA), as well as total mRNA (T-RNA), was obtained from three independent experiments with chondrocytes isolated from three different donors and prepared for high-throughput sequencing.

The expression of 352 genes was significantly changed (FDR < 0.05) following miR-145 overexpression (T-RNA, Appendix A). Out of these genes, only 80 were downregulated (DownT-RNA, Table 1) suggesting that most of the effects of miR-145 on gene expression are indirect. Ninety-six genes were enriched in IP-RNA from samples overexpressing miR-145. Out of these genes, 59 (Table 2) were simultaneously reduced or unchanged in the corresponding T-RNA samples. Since miRNAs normally act by decreasing target-mRNA stability or translation efficiency we considered those 59 mRNAs as miR-145 direct targets (Di-IP, Table 2).

Although miRNAs can also mediate repression through “seedless” sites [29] and/or through the CDS [30], miR-145 direct targets are generally expected to contain miR-145 seed matches in the 3′UTRs. TargetScan (TargetScan (v6.2) [31] http://www.targetscan.org/) detected miR-145 target sites in 17 out of our 59 candidates (Table 2) whereas RNAhybrid, a bioinformatic tool that gives little weight to the seed region and allows the analysis of any given mRNA sequence, including CDSs or 3′UTRs, identified miR-145 binding sites within the 3′UTR, the CDS, and both the CDS and 3′UTR of 8, 20, and 12 candidate genes, respectively (Table 2). We could not detect any predicted binding sites in 12 out of the 59 candidate genes (Table 2). Additionally, we used cWords [28] to analyze words enriched in 3′UTRs and CDSs of IP-RNAs (Figure 2). As expected, the word with the strongest enrichment in CDSs of upregulated genes corresponds to the 6mer-seed site of miR-145 (ACTGGA—Figure 2B) and several shifted variants of the seed site were also enriched [32]. Surprisingly, words corresponding to miR-145 seeds were not significantly enriched in 3′UTRs (Figure 2C,D). Altogether, our data suggest that genes upregulated in IP-RNA (Di-IP) are miR-145 direct targets and that miR-145-target interactions occur predominantly through their CDS.

Further supporting that most of the changes observed in gene expression following miR-145 modulation are indirect, cWords failed to detect an enrichment in miR-145 seed-matching sequences in genes upregulated or downregulated at the mRNA level (data not shown). Moreover, out of the 80 genes in which expression was downregulated by the action of miR-145 (DownT-RNA), only one was included in the Di-IP group. Due to the complexity of the immunoprecipitation protocol and the low amount of mRNA recovered, we could not rule out the possibility that the miR-145–mRNA weak interactions were lost. Indeed, clustering and PCA plots (Appendix A) suggested one potential outlier (IP-RNA, sample corresponding to miR-145 overexpression, donor 3). Thus, we expanded the analysis excluding the outlier (donor 3) and generated an additional list of target candidates (Appendix A), with genes that were significantly enriched in IP-RNA from only two patients (1 and 2) and simultaneously reduced or unchanged in the T-RNA samples (donors 1–3). This additional list contained several genes, which expression was repressed by miR-145 action at the RNA level (DownT-RNA, Table 1, genes marked with *), and the gene Sox9, which we have previously demonstrated to be a direct miR-145 target, regulated exclusively at the protein level in HACs [15]. Consistently, words corresponding to the miR-145 seed were clearly enriched in both CDSs and 3′UTRs of these genes (Appendix A).

As mentioned above, our original RIP-seq results (with three HACs donors) suggest that all the direct target genes identified (Di-IP) but one are repressed by miR-145 at the level of translation, since no significant reduction in their mRNAs was observed upon miR-145 overexpression. Thus, to further validate our findings we set to confirm an effect of miR-145 at the protein level of 12 randomly selected genes from this list. Accordingly, we generated luciferase reporters containing the 3′UTR (CYR61, HSBP1, RTKN, GLT8D1, and PLOD1, with predicted sites only in the 3′UTR, Table 2), the CDS (FKBP2, CTBP1, and FN3K, with predicted sites only in the CDS, Table 2) or both (MOK, RAB7A, and SEC11A with predicted sites in both 3′UTR and CDS and HIST1H4E, lacking predicted sites) of these mRNAs downstream of the firefly luciferase ORF.

Supporting silencing at the protein level, miR-145 overexpression significantly reduced firefly luciferase activity (Figure 3) in the presence of CYR61, HSBP1, GLT8D1, RTKN and PLOD1 3′UTRs, FN3K, MOK and RAB7A CDS, HIST1H4E full-length sequence and SEC11A both 3′UTR and CDS. Only two of the sequences tested (CTBP1 and FKBP2 CDS) failed to mediate silencing. Interestingly, no target sites were predicted for HIST1H4E mRNA, but miR-145 overexpression significantly reduced luciferase expression in the presence of full-length HIST1H4E cDNA, suggesting that HIST1H4E is targeted by miR-145 and demonstrating that our approach can detect targets that would not be predicted by most bioinformatics tools.

It has been previously suggested that miRNA action on translational repression precedes mRNA degradation [33]. Therefore, we hypothesize that miR-145 action in some of our newly identified targets may eventually result in mRNA degradation. Thus, we set to quantify mRNA expression of 16 randomly selected genes from the Di-IP list (Table 2) in HACs transfected with miR-145 mimics for 48 h (instead of 24 h). HACs become gradually dedifferentiated with passage and, therefore, we decided to use freshly isolated (fully differentiated, P0) HACs for these experiments. Accordingly, we overexpressed miR-145 or a control mimic in P0 HACs isolated from seven different patients and analyzed the expression of the selected 16 miR-145 direct targets by qPCR (Figure 4A). Even though their mRNA levels remained unchanged 24 h after transfection (Table 1, Appendix A), four genes were significantly (*p* < 0.5) downregulated and three showed a tendency to be downregulated (*p* < 0.1) 48 h after transfection. As expected, CC2DB1 mRNA, which was the only direct miR-145 target which mRNA was significantly downregulated at the mRNA level 24 h after transfection, was also significantly reduced at the 48 h mark. These data suggest that the miR-145-target interaction in HACs results first in translational repression and it is later on followed by mRNA destabilization.

We also validated the effect of miR-145 in three of the genes present in DownT-RNA but not in Di-IP, showing a significant effect for FSCN1 and UXS1, but not PPP3CA (Figure 4B). We additionally corroborated the miR-145 effect in COL2A1, which we have previously shown [15] to be indirectly regulated by miR-145 (Figure 4B).

In order to identify the biological processes and pathways in which miR-145 is involved, we submitted the full list of genes altered upon miR-145 overexpression (Appendix A, T-RNA FDR < 0.05) together with those included in the Di-IP list (Table 2), to the Database for Annotation, Visualization and Integrated Discovery (DAVID v 6.7). Significantly enriched pathways included immune response and proteolytic processes, both characteristic of cartilage disease (Table 3) (*p* < 0.1).

One of the genes involved in immunity, DUSP6, encodes a dual-specificity phosphatase, MAP kinase phosphatase 3 (MKP3) that inhibits MAPK activity by dephosphorylating threonine and serine residues. MKP3 regulates the elongation of cartilage elements of digit 2 during development through repression of FGF signaling [34]. Interestingly, loss of DUSP6 in mouse embryos leads to dominant, incompletely penetrant, and variable phenotypes including hearing loss, coronal craniosynostosis, and skeletal dwarfism. The latter phenotype has also been observed upon mutations in Sox9 [35]. In addition, chondrocytes in the mutant’s growth plate-proliferating zone were disorganized [36]. Thus, we hypothesized that miR-145-mediated DUSP6 regulation affects adult chondrocyte function. As expected, miR-145 overexpression in HACs led to a strong reduction of DUSP6 protein (Figure 5A,B) in both regular (20%) and low (1%, hypoxia) oxygen tension, which is the normal physiologic condition for HACs and promotes their differentiated phenotype. Conversely, prevention of the miR-145 function with specific inhibitors resulted in increased DUSP6 expression (Figure 5C). As previously shown, miR-145 overexpression causes only a small [35], not significant, decrease in DUSP6 mRNA (Figure 4A), suggesting that its effect occurs mainly at the translational level. A miR-145 binding site was predicted in the position 1018–1025 of DUSP6 3′UTR. In order to validate the functionality of this binding site, we generated reporters containing the wild type DUSP6 3′UTR or a mutated version, with point mutations in three nucleotides of the target site, downstream of the luciferase ORF. Transfection of miR-145 mimics significantly reduced luciferase activity of the wild type construct but failed to silence luciferase activity in the presence of the mutations (Figure 5D). Our data confirm that miR-145 represses DUSP6 expression through the predicted binding site in its 3′UTR.

FGF2 (fibroblast growth factor 2) plays an anti-anabolic and/or catabolic role in HACS [37,38] at least partially via the activation of the Ras-Raf-MEK1/2-ERK1/2 pathways [39]. FGF2 is present in OA synovial fluid, where it activates Runx2 and promotes MMP13 expression [3]. Levels of FGF2 and MMP13 are increased in human OA cartilage and, in adult human articular cartilage, FGF2 stimulates MMP13 through the activation of multiple MAPKKs [40]. Additionally, mice in which increased Erk1/2 phosphorilation occurred upon heparin endosulfatases (Sulf1 and Sulf2) depletion, showed increased MMP13 levels and reduced Col2a1 and ACAN [41] expression. Interestingly, a raise in Runx2 and MMP13 and a reduction in Col2a1 and ACAN was observed upon miR-145 overexpression ^15^.

In mouse embryos, transcription of DUSP6 is dependent on FGF signaling and targeted inactivation of DUPS6 increased the levels of pERK, the main target described for DUSP6 [36]. Thus, we hypothesized that DUSP6 contributes to the effect of miR-145 in the expression of the catabolic/anabolic molecules mentioned above. Whereas DUSP6 depletion with specific siRNAs did not alter Col2a1 or ACAN levels (data not shown), expression of MMP13 was strongly upregulated (Figure 6A), indicating that miR-145-dependent MMP13 regulation may be partially mediated by DUSP6. Nevertheless, miR-145 overexpression or siRNA-mediated DUSP6 depletion failed to impair ERK phosphorylation in response to FGF2 (Figure 6B,C). Altogether, these data indicate that miR-145 function can be modified without significantly interfering with FGF2-dependant ERK activation and the effect of DUSP6 in MMP13 expression occurs through a still unidentified ERK1/2-independent pathway.

## 4. Discussion

We have applied RIP-seq in combination with total mRNA-seq following miR-145 overexpression to successfully identify > 50 new direct miR-145 targets in primary human articular chondrocytes. MiRNA-target interaction results in translational repression accompanied or not of mRNA degradation [42,43,44]. Therefore, experimental approaches that rely on transcriptome profiling alone fail to detect targets regulated exclusively at the level of translation or to differentiate between direct and indirect targets. Surprisingly, most miR-145-mRNA interactions in HACs did not result in mRNA degradation 24 h after overexpression. However, a reduction in several of these transcripts was observed at a later time point and luciferase assays further confirmed that miR-145 overexpression results in lower protein (luciferase) levels for the majority of the cases (10 out of 12). Therefore, our results suggest that miR-145 functions by inhibiting translation of most of its direct targets and this is later followed by mRNA decay. The main mechanism by which miRNAs repress gene expression remains highly controversial [42,45,46]. Interestingly, recent studies suggest that the mechanism of action might be cell-type specific [47] and dependent on the target site type. Thus, Liu et al. found that extensive complementarity to the seed region of the miRNA through the 3′UTR often results in mRNA decay instead of translational repression [43]. Nevertheless, it remains unclear which features favor mRNA translation over decay [43]. We demonstrate here that, in HACs, miR-145 binding occurs more often through CDSs. Recent studies have shown that interaction through the CDS occurs frequently [22,30,48]. Whether the region of interaction miR-145/mRNA influences the mechanism of silencing in HACs remains to be clarified. While most identified miR-145 targets contained predicted miR-145 binding sites for the miRNA, 12 genes (20%) were not predicted as miR-145 targets by TargetScan or RNAhybrid. One of these genes, HIST1H4E, was within the list of targets selected randomly for further validation. Overexpression of miR-145 reduced luciferase activity in the presence of the full HIST1H4E mRNA, suggesting that miR-145 acts on this sequence to repress translation, possibly via a noncanonical binding site. MiRNA action through noncanonical and seed-independent binding sites is well documented [49,50].

Even though our RIP-seq and luciferase reporter results suggest that our list of direct targets genes (Table 2) are silenced by direct binding of miR-145 to their mRNAs, similar results could potentially be obtained if miR-145 overexpression resulted in altered expression of other miRNAs. Luciferase assays with reporters containing mutations in the predicted miR-145 binding sites would further confirm a miR-145-target mRNA direct interaction. While we have performed these experiments for DUSP6, further mutation assays are outside the scope of this work. Moreover, whereas we have previously observed SOX9-dependent changes in miR-140 and miR-365 [15] expression 48 h after miR-145 overexpression, the RIP-seq and reporter experiments presented here have been performed within 24 h of miR-145 overexpression, to limit the occurrence of indirect effects.

Importantly, computational tools for miRNA target predictions typically observe hundreds to thousands of targets (905 transcripts contain conserved miR-145 binding sites according to TargetScan), including a high number of false-positive predictions [18]. Our approach allowed us to identify miR-145 targets in a cell-specific manner, reducing the number of false positives.

MiR-145 overexpression resulted in upregulation of a high number of genes (272—Appendix A) and downregulation of > 80 genes (Table 1 and Appendix A), suggesting indirect effects. This is not entirely surprising, since, as we have shown in the past, an important target of miR-145 is SOX9 [15], a transcription factor that regulates a wide range of genes in HACs. Further supporting this idea, only one of the downregulated genes was enriched in Ago-IPs although several additional genes from this list were enriched in Ago-IPs upon exclusion from the analysis of one outlier sample. Thus, our immunoprecipitation protocol may have failed to recover weak mRNA–miRISC interactions. mRNA crosslinking may increase the yield of recovered interactions, although it could lead to potential artifacts due to reduced cell lysis efficiency, introduce sequence biases, increase background, or be incompletely reversible [51,52]. Increasing the number of RIP-seq experiments would have definitely resulted in a more comprehensive list of actual targets, but this was limited both by the availability of healthy primary chondrocytes and the high cost of the HT-seq, rapidly decreasing as the technology is optimized. Importantly, mRNA-RISC is a dynamic interaction, which can be hugely affected by the cellular context: Our experiments have been performed in primary human chondrocytes, instead in the more accessible cell lines or mouse models that are commonly used in miRNA studies in cartilage, increasing the reliability and applicability of our results.

Although it remains to be demonstrated, several of the newly identified miR-145 targets in HACs may play important roles in cartilage. For example, POLDIP2 may modulate the production of MMP1 through NOX4 [53,54]. Moreover, PLOD1 could potentially be involved in collagen metabolism [55] and ERF, an ETS2 repressor factor, may indirectly modulate the expression of ECM components [56,57]. Interestingly, CYR61 has already been shown to play a role in chondrogenesis [58], although its function in adult cartilage remains to be clarified.

MiR-145 is abundantly expressed in cartilage, but is also highly expressed in various mouse tissues such as aorta, where miR-145 affects its structure [59]. Moreover, changes in miR-145 levels have been associated with various human diseases, including stroke [60], arteriosclerosis [61], or cancer [62,63]. Although miRNA-target interactions can be cell-specific, our list of targets in HACs could be of interest to researchers in other fields and some of the interactions may be relevant for other tissues or diseases. Supporting this idea, RTKN has been already proven to be targeted by miR-145 in breast cancer [62], FSNC1 in colon cancer and melanoma [64,65], and GOLM1 in prostate cancer [66].

A growing number of studies have associated changes in the expression of dozens of miRNAs to the development of OA [67]. Well studied examples include protective miR-140 [68] or detrimental miR-138 [69]. Moreover, miRNAs have also been proposed as biomarkers of the disease [67] that could facilitate early OA detection. This is important because symptoms are often shown once the cartilage is too degraded for intrinsic repair. Nevertheless, the specific function and gene targets of most of these miRNAs, and thus whether OA-related changes contribute or are consequential to the disease, remains unknown. The experimental approach used here can be applied to the study of any other chondrocyte miRNA and will serve to understand whether a given miRNA has therapeutic potential for OA.

OA pathology is characterized by progressive cellular and molecular changes in joint tissues. While it has always been considered a degenerative joint disease influenced by mechanical stresses and ageing causing cartilage breakdown, discoveries in the recent years point towards a critical contribution of the immune system and inflammatory mechanisms [2,70]. Chondrocytes in the joint can express and respond to cytokines and chemokines such as TNF-a, IL-1 or IL-6, and both IL-1β and Interferon-γ suppress COL2A1 transcription. Moreover, inflammatory signals can contribute to macrophage activation within the osteoarthritic joint, which can themselves produce cytokines such as IL-1 or TNFα [2]. Interestingly, Yang et al. [16] have recently found that miR-145 is upregulated in OA cartilage and chondrocytes, whereas HACs treated with IL-1β contained higher miR-145 levels. Moreover, inhibition of miR-145 partially reversed the IL-1β-mediated reduction in COL2A1 and ACAN as well as the stimulation of MMP-13. On the contrary, salidroside (SAL), a compound with pharmacological effects suggested to accelerate fracture healing [71], has been recently shown to alleviate LPS-mediated inflammatory injury in a chondrogenic cell line by, at least partially, stimulating miR-145 expression [72]. Our gene ontology analysis of the genes affected by overexpression of miR-145 show enrichment in pathways involved in immunity and inflammatory and interferon responses. Not surprisingly, a role for miR-145 in modulating interferon response has already been suggested in very different cellular contexts such as bladder cancer [73] or fish INF-γ-induced immune response [74]. Nevertheless, the mechanisms underlying the role of miR-145 in these pathways remain unknown. Whereas miR-145 induces INF-β by targeting the suppressor of cytokine signaling 7 (SOCS7), we did not detect an effect of miR-145 in SOCS7 expression or association with miRISC. Thus, miR-145 most probably modulates the inflammatory response in HACs through a SOCS7-independent mechanism. From our list of miR-145 targets in HACs, DUSP6 stood up as an interesting candidate to control inflammatory responses in HACs. DUSP6 has been previously shown to mediate the response of macrophages to LPS [75] and plays dual roles in the regulation of vascular inflammation [76]. Moreover, DUSP6 acts on FGF signaling to control cartilage digit development [34] and growth-plate chondrocytes organization [36] during mice development. Our work demonstrates that miR-145 silences DUSP6 expression at the protein level in primary adult HACs. In adult HACS, FGF2-ERK activation promotes the expression of MMP13 and other catabolic molecules [37,38], which are often increased in OA. Here, we show that the effect of miR-145 in MMP13 expression, previously described by Yang [16] and ourselves [15] might be at least partially mediated by DUSP6. Nevertheless, in our hands, the levels of pERK upon FGF2 stimulation were similar in the absence of DUSP6 and/or presence of miR-145, suggesting that the effect of DUSP6 on MMP13 is independent of FGF2-mediated ERK phosphorylation. Further studies, especially those mimicking cartilage diseases, are required to confirm these findings and to shed more light into the mechanism of action of this phosphatase in HACs.

In summary, we demonstrate that miR-145 directly targets dozens of genes in the human articular chondrocyte to strongly impact its function. We have previously proposed that miR-145 inhibition in OA cartilage results in increased synthesis of key ECM components, such as COL2A1, and silencing of degradative enzymes through its target *SOX9* [22]. Our new findings support the argument that OA cartilage treated with miR-145 inhibitors would additionally benefit from a reduction in the inflammatory responses that contribute to cartilage degeneration. MiRNAs are hot candidates as drugs/drug targets and promising advances on the delivery methods and stability of both miRNA mimics and inhibitors (often referred to as antagoMiRs) have been made during the past decade [77]. Articular cartilage is poorly vascularized, alymphatic and aneural, and cartilage cells are embedded in a thick ECM, impairing effective pharmacologic intervention. Thus, autologous chondrocyte implantation has been suggested as a promising approach for cartilage repair while MSCs have been proposed as an exciting source of chondrocytes for autologous or allogenic transplantation. Suppression of miR-145 has been shown to promote chondrogenic differentiation of MSCs [78] in vitro. Our results suggest that inhibition of miR-145 could also prevent hypertrophy of the transplanted tissue. The effect of modulating the expression of miR-145 target genes for chondrocyte function will need to be carefully assessed prior to the development of therapeutic approaches targeting miR-145. Experiments performed in vivo will surely address the suitability of miR-145-based therapies for the treatment of OA in the near future.

## Figures and Tables

**Figure 1 life-10-00058-f001:**
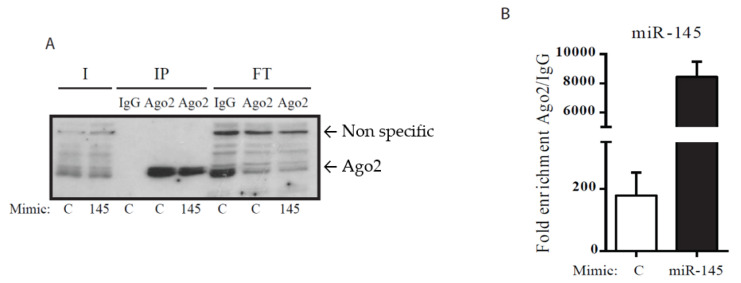
Immunoblotting analysis of human articular chondrocyte (HAC) lysates following transfection with miR-145 mimics. HACs were transfected with control (C) or miR-145 (145) mimics and immunoprecipitation was performed with an anti-Ago2 antibody (Ago2) or an IgG control, as indicated. I (Input): 2.5% of the total lysate submitted to immunoprecipitation (IP); FT (Flow-through): 5% of the extract recovered after IP. (**A**) Samples were separated by SDS-PAGE and proven with the anti-Ago2 antibody to determine the presence of Ago proteins. (**B**) MiR-145 bound to Ago2 following immunoprecipitation, in comparison with IgG, was estimated by RT-qPCR. MiR-145 overexpression resulted in ~40 fold increased in the amount of miR-145 bound to Ago2.

**Figure 2 life-10-00058-f002:**
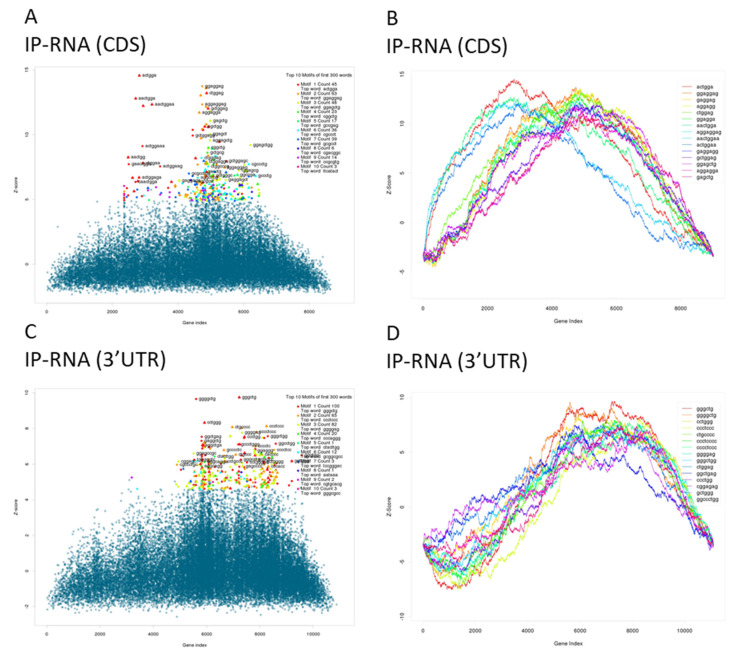
Enrichment of miR-145 seed sites in the CDSs (**A**,**B**) or 3′UTRs (**C**,**D**) of the precipitated mRNAs. (**A**,**C**) Scatter plot showing the maxima of enrichment profile. Each dot represents a word, the Y-position reflects the maximum score of an enriched word and X-position shows the gene-index where the Z-score is maximum. Top-ranked words with a left-shift associate with gene-expression change (lower gene index represents a stronger upregulation in Ago2 upon miR-145 transfection). Triangles annotate known seed sites of human miRNAs, red triangles show miR-145 seed sites. (**B**,**D**) Word-enrichment profile: The line plot shows enrichment through the gene rank for the top ten enriched words. Each line represents the running sum over all scores that quantify a degree of enrichment according to gene upregulation (from most upregulated to most downregulated).

**Figure 3 life-10-00058-f003:**
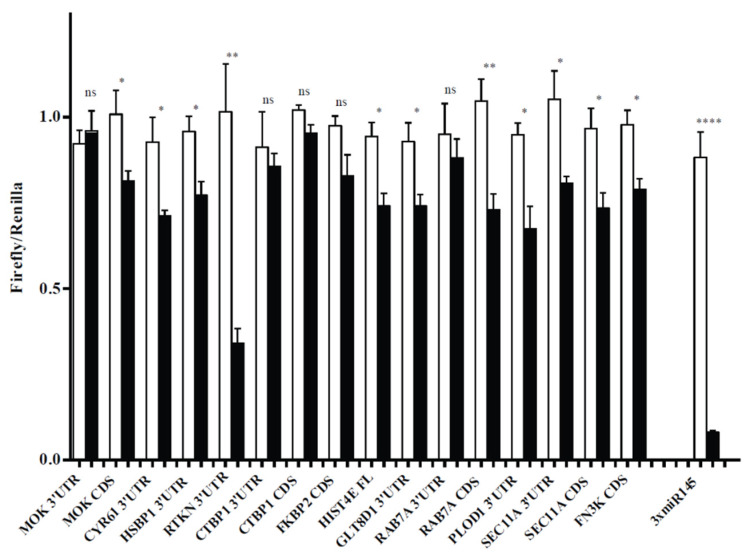
Luciferase reporter assays to validate a miR-145 direct effect on the expression of selected genes. Hela cells were transfected with luciferase reporters containing full-length CDSs or 3′UTRs, or both, of 12 randomly selected mRNAs from Di-IP, as indicated, downstream the firefly luciferase ORF. Control (white bars) or miR-145 (black bars) mimics were cotransfected in the cells. Values are normalized to the levels of renilla luciferase, independently expressed by the same vector and are shown as relative to that obtained for each construct cotransfected with the control miRNA mimic (± SEM). As positive control, a construct containing three perfectly complementary binding sites for miR-145 (3 x miR-145) was included in the experiment. * *p* < 0.05; **** *p* < 0.0001; ns: Not significant.

**Figure 4 life-10-00058-f004:**
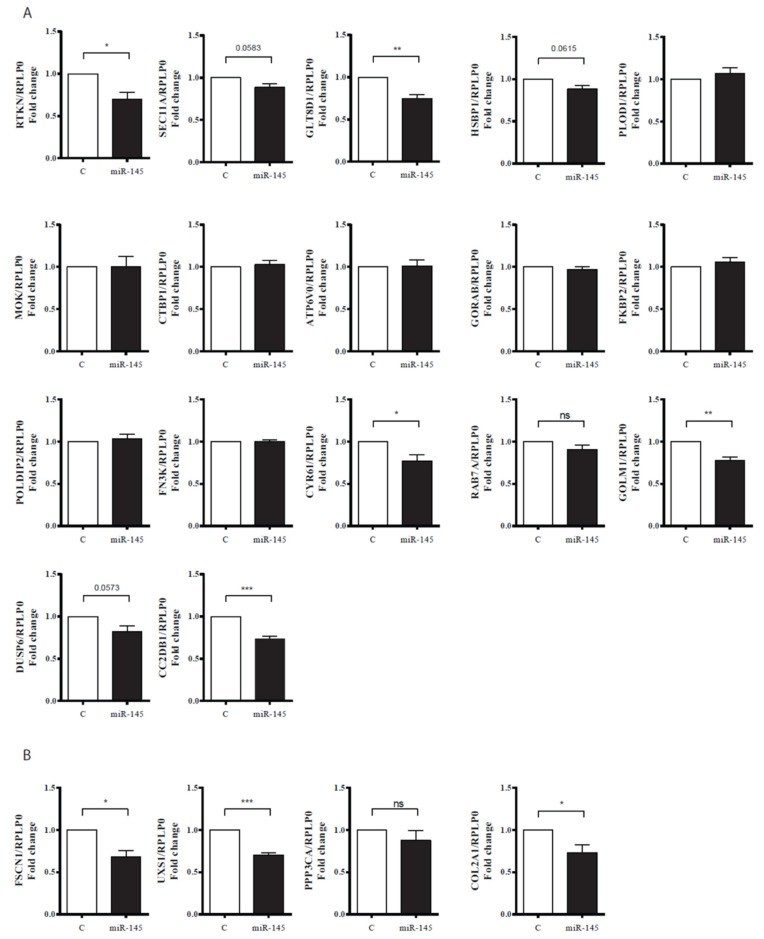
Analysis of the effect of miR-145 overexpression on mRNA levels of selected genes in freshly isolated HACs. (**A**) Sixteen randomly selected mRNAs from Di-IP and CD2DB1 from both Di-IP and DownT-RNA. (**B**) Three genes only present in DownT-RNA and COL2A1. Control (C) or miR-145 mimics were transfected in freshly isolated HACs and RNA was extracted 48 h after transfection. Values are presented as relative to that obtained in cells transfected with control mimics for each patient and normalized to RPLP0. Data represents average ± SEM from seven different experiments, each performed with different donor cells (19 yo. female; 8 yo. male; 28 yo. male; 45 yo. male; 16 yo. male; 13 yo. female; 31 yo. female). * *p* < 0.05; ** *p* < 0.01; *** *p* < 0.001; ns: Not significant.

**Figure 5 life-10-00058-f005:**
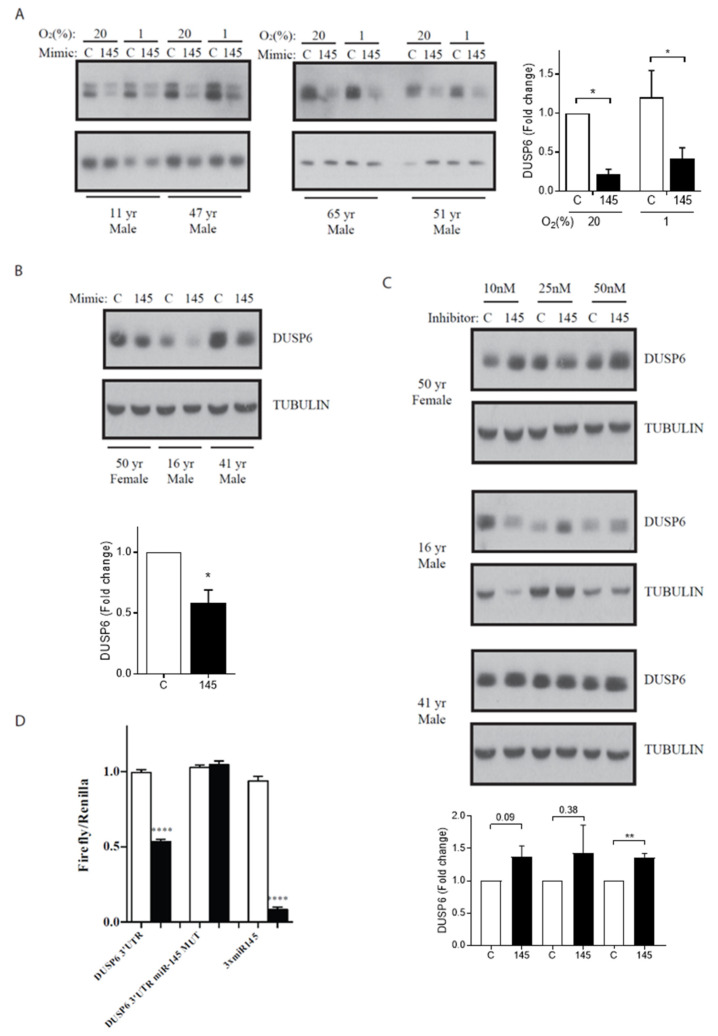
MiR-145 directly represses DUSP6. Western blot showing reduced DUSP6 levels following miR-145 overexpression in P2 (**A**) and freshly isolated (**B**) HACs. **(C)** Western blot showing increased DUSP6 levels following miR-145 inhibition in freshly isolated HACs. DUSP6 protein bands were quantified by densitometry and are normalized against tubulin and expressed relative to the control (± SEM). * *p* < 0.05, ** *p* < 0.01 (A, right hand side panel, B and C, lower panels). In all experiments HACs were transfected with a relevant control (C) or miR-145 precursor or inhibitor, and subsequently cultured in 20% or, where indicated, 1% O2 tension for 44 h. **(****D)** Hela cells were transfected with luciferase reporters containing three perfectly complementary binding sites for miR-145 (3 x miR-145), the DUSP6 3′UTR containing a putative miR-145 binding site (seed matching nucleotides 1018 to 1025; DUSP6 3′UTR), or a mutated seed-matching site (DUSP6 3′UTR miR-145 MUT) downstream the firefly luciferase open reading frame (ORF). Control (‘C’) or miR-145 mimics were cotransfected in the cells. Values were normalized to the levels of renilla luciferase, independently expressed by the same vector and are shown as relative to that obtained for each construct cotransfected with the control miRNA mimic (± SEM). **** *p* < 0.0001.

**Figure 6 life-10-00058-f006:**
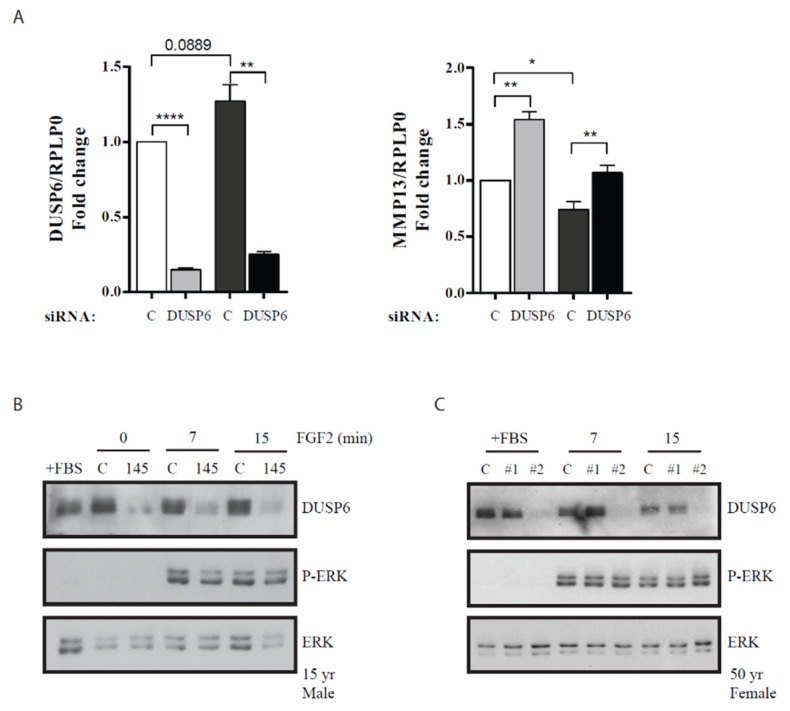
DUSP6 depletion results in MMP13 upregulation. (**A**) RT-qPCR analysis of DUSP6 and MMP13 levels after transfection of P2 HACs with 5 nM DUSP6 siRNA. Values are presented as relative to that obtained in cells transfected with control siRNA for each patient and normalized to RPLP0. Data represents average ± SEM from four different experiments, each performed with different donor cells (11 yo. male; 47 yo. male; 65 yo. male; 51 yo. male). * *p* < 0.05; ** *p* < 0.01; **** *p* < 0.0001. (**B**,**C**) Western blot showing ERK phosphorylation (pERK) after treatment with FGF2 (25 nM). Total ERK (ERK) is shown as a loading control. P1 HACs were transfected with control (C) or miR-145 (145) mimics (**B**), or control (C) or DUSP6 siRNAs (#1, #2, notice than only #2 siRNA results in DUSP6 depletion, therefore #1 acts as an additional control) (**C**). After 24 h, HACs were placed in a serum-free medium overnight and stimulated with 25 nM FGF2 for the indicated times. These experiments were performed three times, with three different donors; one representative experiment is shown.

**Table 1 life-10-00058-t001:** MicroRNAs (mRNAs) significantly downregulated (False Discovery Rate (FDR) < 0.05) upon miR-145 overexpression (DownT-RNA: Total downregulated mRNA). Control or miR-145 mimics were transfected in P3 HACs from three different patients. After transfection by 24 h, cells were lysed and RNA was extracted and submitted to high-throughput (HT)-seq. **: Genes significantly enriched in Di-IP; *: Genes significantly enriched in Ago2-IP following miR-145 overexpression when the outlier patient was excluded from the analysis (Appendix A).

Gene	LogFC	FDR
RPL3L	−2.31149	0.018536
RP11-365D23.4	−2.16109	0.02382
FSCN1 *	−1.44751	1.52 × 10^−10^
HIST1H4L	−1.39957	0.013354
RP11-498C9.15	−1.39914	0.007266
STOX2	−1.3254	0.034993
RP11-649A18.7	−1.02532	0.024884
PKP1	−1.00064	0.032246
ABRACL	−0.98904	4.36 × 10^−7^
CXCL6	−0.95445	0.012477
RP11-166P13.3	−0.93763	0.024801
CTC-448F2.1	−0.88035	0.000899
MALAT1	−0.85554	0.00094
EID2B	−0.84762	0.02382
DTD1	−0.81024	0.000533
ADPGK	−0.78387	0.002089
CLCN2	−0.7662	0.017386
AC145291.1	−0.75286	0.010585
RP11-620J15.3	−0.75057	0.000152
FAM108C1 *	−0.72829	0.011192
CXCL1	−0.72783	0.010556
TMOD2	−0.72433	0.011264
SERPINE1	−0.72202	0.00137
TMEM9B	−0.72166	0.000757
USP46	−0.71885	0.004563
GCSH	−0.70966	0.001641
NT5DC2	−0.70882	0.001507
GMFB *	−0.68607	0.004577
AL132709.8	−0.68606	0.008869
HOMER2	−0.68387	0.001653
DANCR	−0.68207	0.015835
CTC-459F4.2	−0.68075	0.034195
ZMYND10	−0.6622	0.039097
MTRNR2L10	−0.65594	0.028528
CCDC43 *	−0.65181	0.004142
RPL13AP25	−0.65105	0.039418
AKR1B1	−0.65088	0.001828
SNHG9	−0.64141	0.01704
BAALC	−0.64103	0.011259
UXS1	−0.63656	0.002733
AC005035.1	−0.63071	0.025323
PIGF *	−0.6181	0.045224
VASN	−0.61722	0.000629
RP1-278E11.3	−0.60875	0.004343
AKR1B10	−0.60848	0.049377
ATP13A2	−0.60351	0.002099
SNX24	−0.5985	0.001368
ZNRD1	−0.59494	0.014698
AL132709.5	−0.5915	0.001592
GPRC5C	−0.58401	0.048364
C15orf59	−0.58256	0.014698
GCLM	−0.55711	0.008631
CC2D1B **	−0.55688	0.012839
PCBP4	−0.5459	0.037638
CTB-63M22.1	−0.53644	0.008388
PPP3CA	−0.53575	0.007346
PMEPA1	−0.5323	0.028545
TRIM65	−0.52934	0.04805
FBF1	−0.52885	0.039097
G0S2	−0.52582	0.045454
FAM45A	−0.52388	0.016243
RPL22L1	−0.51543	0.02124
MAP3K11	−0.51431	0.017156
RPS3AP5	−0.49364	0.009962
MT1E	−0.48805	0.02457
PSAT1	−0.48716	0.01541
SMO	−0.48585	0.049462
SWI5	−0.48225	0.028528
H2AFX	−0.47418	0.01541
PAFAH1B2	−0.46644	0.037335
DTYMK	−0.46583	0.042746
RP11-543P15.1	−0.4624	0.027114
IMP3 *	−0.462	0.04281
RPS28	−0.46182	0.013354
RP11-20O24.4	−0.44976	0.034761
RP1-292B18.1	−0.44362	0.041162
RPS2P55	−0.43723	0.037335
RP11-864N7.2	−0.43695	0.040095
RPL9	−0.43489	0.044893
C7orf73 *	−0.42472	0.042746

**Table 2 life-10-00058-t002:** Significantly enriched mRNAs (FDR < 0.05) in Ago2 immunoprecipitates following miR-145 overexpression (Di-IP: Direct targets (candidates)). Control or miR-145 mimics were transfected in P3 HACs from three different patients. After transfection by 24 h, cells were lysed and submitted to immunoprecipitation with anti-Ago2 antibodies. Immunoprecipitated RNA was extracted and submitted to HT-seq. The list only includes those genes, which expression was downregulated (*) or unchanged as a consequence of miR-145 overexpression. The last three columns indicate the presence of predicted binding sites (+) accordingly to TargetScan (3′ untranslated region (3′UTRs)) and RNAhybrid (3′UTRs or coding sequences (CDSs), as indicated).

Gene	LogFC	FDR	TargetScan	RNAhybrid (3’UTR)	RNAhybrid (CDS)
COA6 (C1orf31)	4.783081	4.8 × 10^−6^			+
EXOG	3.41819	9.44 × 10^−5^			
HIST1H4E	3.404645	4.49 × 10^−19^			
RP11-539I5.1	2.924915	0.0196			
HSBP1	2.435244	3.97 × 10^−9^	+		
HIRIP3	2.303426	0.005801			+
DCAF13	2.288777	2.13 × 10^−6^			
BANF1	2.250307	3.08 × 10^−12^			+
RAB7A	2.223753	5.22 × 10^−5^			+
TMEM223	2.20113	0.007146			+
FN3K	2.177953	0.001289			+
SEC11A	2.040026	9.75 × 10^−10^		+	+
MOK	2.005743	0.000539		+	+
TALDO1	1.989063	1.19 × 10^−9^			+
ULBP3	1.963432	0.032833			+
ARHGEF10L	1.93225	2.92 × 10^−8^		+	+
CYR61	1.868992	1.77 × 10^−6^	+	+	
TMOD3	1.818792	0.001511	+	+	
GORAB	1.803528	0.030719	+	+	
C6orf106	1.772683	1.55 × 10^−7^		+	
SUPT16H	1.764385	0.004327			
DECR1	1.743822	0.029152			
CC2D1B*	1.743593	3.69 × 10^−6^	+	+	+
FARSA	1.737866	5.11 × 10^−7^			+
TPI1P1	1.723506	5.57 × 10^−7^			
GLT8D1	1.698869	0.000477	+		
MRPL51	1.659022	1.95 × 10^−5^			+
ATP6V0B	1.653234	0.000656	+	+	
FAM96A	1.632677	0.023186			
FKBP2	1.60429	0.001637			+
SCAND1	1.593582	0.014036			
IFFO1	1.551748	0.001511		+	+
CDC6	1.543387	0.012934			+
PSMD11	1.532049	0.007204	+	+	+
MCM5	1.531026	0.013155			+
MBD2	1.495552	0.013169	+		+
MRPL41	1.473089	0.029152			+
TPI1	1.451595	0.005256		+	+
DUSP6	1.424208	0.019572	+	+	
TIGD5	1.419209	0.007146		+	+
RPL39	1.411136	0.025767			
OGFR	1.399202	0.002343			+
RTKN	1.364258	0.04595	+	+	+
CTBP1	1.351367	0.00019			+
ARL2	1.332863	0.002363			+
TADA3	1.239423	0.032833			+
PTGR1	1.236809	0.016147			
KLHDC10	1.192891	0.048934	+	+	+
GOLM1	1.141642	0.006373	+		+
METRNL	1.100353	0.015776			+
PLOD1	1.098426	0.047772	+		+
ADM	1.08839	0.012934			
SIX1	1.066897	0.042474	+	+	
RCN3	1.046073	0.038849			+
POLDIP2	1.041776	0.015349		+	
COX6B1	1.032669	0.030567			+
KIAA0930	1.010146	0.030719	+	+	+
ERF	1.006507	0.047544	+	+	+
ATP5A1	0.984198	0.029152			+

**Table 3 life-10-00058-t003:** Enriched Gene Ontology (GO) Terms. Genes altered at the mRNA level upon miR-145 overexpression (FDR < 0.05, Appendix A) and direct miR-145 mRNA targets (Table 2) were submitted to the Database for Annotation, Visualization and Integrated Discovery (DAVID v 6.7) for enrichment analysis of biological processes (classification category Panther Biological Processes (Panther_BP_All)).

Term	*p*-Value
Interferon-mediated immunity	1.10 × 10^−13^
Immunity and defense	3.30 × 10^−5^
Proteolysis	8.80 × 10^−3^
Glycolysis	1.20 × 10^−2^
T-cell mediated immunity	1.30 × 10^−2^
Macrophage-mediated immunity	2.30 × 10^−2^
MHCI-mediated immunity	4.90 × 10^−2^
Protein metabolism and modification	6.50 × 10^−2^
Protein biosynthesis	9.00 × 10^−2^
AngiogenesisCOA6 (C1orf31)	9.60 × 10^−2^

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
