# Peer review of "High-Throughput Identification of MiR-145 Targets in Human Articular Chondrocytes"

_life, 2020, doi:10.3390/life10050058_

Round 1

Reviewer 1 Report

The authors of manuscript “High through-put identification of miR-145 targets in human articular chondrocytes” studied the role of miR-145 as a regulator of several genes, in particular genes involved in extracellular matrix and inflammation, in primary chondrocytes.

In this study, the authors concluded that miR-145 is an important regulator of chondrocyte function and a new possible target for cartilage repair in OA patients.

The idea project's is interesting. Moreover, in my opinion the manuscript can be accepted for the publication in this journal after a major revision.

Concerning this, the manuscript can be improved according to following reviewer suggestions. 

  • In the introduction section I encourage: adding information on the failure of the therapeutic approach of OA , in particular as regards the direct action on chondrocyte cells; to insert a better description of the experimental design. This part is confusing.
  • In the materials and methods section, several details are missing in the description of the methods. For example: used tools, product codes, company details, in which medium the isolated cells are maintained, how the treatment is carried out in different quantities of oxygen, etc.
  • In the results section, I suggest to better explain the data obtained, in particular pages 12-13.
  • In discussion: the data were well described. In my opinion it is better to add the clinical relevance of the data obtained, in order to also attract the attention of clinical readers.
  • I encourage the addition of recent studies regarding the new miRNAs identified as biomarkers or therapeutic targets in OA progression or treatments.
  • I suggest to check english language and style.

Author Response

Thank you very much for your time reviewing this manuscript. Please see the attachment including responses to all reviewers.

Reviewer 2 Report

The manuscript "High through-put identification of miR-145 targets in human articular chondrocytes" is, overall, a scientifically interesting paper that adds knowledge about the role of microRNAs in the regulation of the metabolic activities of articular chondrocytes.

However, the paper should only be considered for publication in Life Journal after major revisions are made.

The Materials and Methods section should be described in more detail.

There is a lack of clarity of the type of subjects (cadavers or fractures?) and which articular joints the healthy chondrocytes are isolated from.

Age and gender of individuals only used to isolate P3 chondrocytes for the Ribonucleoprotein Immunoprecipitation are mentioned in the Materials and Methods section. Maybe details of other subjects used as cell source for other experiments should be added in this section too.

A description of the control mimics and of the DUSP6 siRNAs (Product codes) should be added. The cell transfection procedure and the concentration of the oligonucleotides used are not shown properly.

It would be also useful to specify the details and the concentration of each antibody used for the WB including the secondary antibodies.

There is no mention of the hypoxic culture conditions in the materials and methods paragraph.

The impossibility to add more samples for the HT-Seq is understandable, but more info about the analysis to identify the outlier and the PCA plots would be beneficial to include.

The presentation of the data through figures and tables is very confusing. Some figures/tables are missing (Figure 1A to show the specificity of Ago2 immunoprecipitation by WB, Figure 1B to show mir-145 overexpression, Table 3 to show enriched pathways, Figure 2 related to cWords analysis, such as the supplementary tables). It's not clear if the data shown in the tables have been obtained after the outlier removal. All the figures, and captions, need to be revised.

It would be appropriate to perform luciferase assays for target confirmation with mutated sequences of the target sites to ensure specificity of the sequence recognition.

A further suggestion would be to explain the rationale for choosing the genes to validate as targets. Maybe, a consensus path analysis to cluster them in specific relevant pathways? 

The WB results don't include a densitometric analysis and the last figure doesn't show a loading control.

The discussion about the results of the DUSP6 gene and why it has been selected for further analysis could be more deeply discussed.

I think the running title is not required by the Life journal.

Author Response

(The authors gave the same response as above.)

Reviewer 3 Report

The manuscript "High though-put identification of miR-145 targets in human articular chondrocytes  (running head: miR-145 targets in human chondrocytes), written by Martinez-Sanchez A, Lazzarano S, Sharma E and Murphy CL analyses the influence of miR-145 expression modulation on the gene and protein expression in human articular chondrocytes. The aim was to identify the targetome of miR-145 in human chondrocytes. Complex methods were used, such as RNA immunoprecipitation plus HT sequencing of miRISC in addition to RT-qPCR, Western blot and luciferase assays. It was demonstrated that miR-145 targets DUSP6, involved in cartilage development, and can influence the expression of MMP13.

The manuscript covers an interesting and original topic and is well organized. The results are well presented, the data thoroughly analysed and discussed. The presented data, as experiments were done on primary cultures, could be significant for further investigations in vivo.

I recommend the manuscript for the publication.

Minor observation:

The units should be written separately from the numbers.

Author Response

We thank the reviewer for his/her positive comments on our manuscript and we have revised the manuscript to separate all units from the numbers.

Round 2

Reviewer 1 Report

Dear Authors,

thank you for answered my questions.

Concerning the study, I think that the revised manuscript requires a final check of the English language and style
In my opinion, the manuscript can be accepeted for the publication in this journal.

Author Response

Thank you very much for your time. We are very pleased you found our manuscript good for publication in its revised form. As suggested, we have further revised English style and edited minor spell mistakes in the text (marked in blue). BW.

Reviewer 2 Report

Dear Authors,

thanks for the amendments. They improved the scientific quality of the manuscript.

Author Response

Thank you very much for your time. We are very pleased you found an improvement in the quality of the manuscript after revision. As suggested, we have further revised English style and edited minor spell mistakes in the text (marked in blue). BW.